# LEVERAGING GRAPH NEURAL NETWORKS TO BOOST FINE-GRAINED IMAGE CLASSIFICATION

## ABSTRACT

Fine-grained image classification, which is a challenging task in computer vision, requires precise differentiation among visually similar object categories. In this paper, we introduce a novel approach that utilizes Graph Neural Network (GNN) blocks to enhance the clustering capability of feature vectors extracted from images within a deep neural network (DNN) framework. These GNN blocks capture intricate dependencies between feature vectors by modeling them as nodes within a graph. This graph-based approach enables our model to learn contextual information and relationships that are essential for fine-grained categorization. In practice, our proposed method demonstrates significant improvements in the accuracy of different fine-grained classifiers, with an average increase of $(+2.78\%)$ and $(+3.83\%)$ on the CUB200-2011 and Stanford Dog datasets, respectively, while achieving a state-of-the-art result $(95.79\%)$ on the Stanford Dog dataset. Furthermore, our method serves as a plug-in refinement module and can be easily integrated into different architectures.

## 1 INTRODUCTION

Fine-grained classification is an important task in computer vision. With the rapid advancement of technology, we now have the capability to collect and store a large amount of image data from various sources. However, classifying objects in images with high similarity, such as bird species, types of leaves, or electronic product models, remains a difficult challenge. This challenging problem has numerous real-world applications, including image recognition, disease diagnosis (Lu et al., 2023; Zhang et al., 2023; Wen et al., 2023), and even biodiversity monitoring (Horn et al., 2017; 2015a;b), where distinguishing between visually similar subcategories is crucial. Despite significant progress in using deep neural networks (DNN) to address this issue, there are still many challenges to overcome in order to achieve high accuracy and stability.

In contrast to standard image classification, fine-grained image classification presents greater difficulty for three primary reasons: (i) substantial intra-class variation, with objects in the same category exhibiting significant pose and viewpoint differences; (ii) subtle inter-class distinctions, where objects from different categories may closely resemble each other with minor differences; (iii) constraints on training data, as labeling fine-grained categories often demands specialized expertise and a substantial amount of annotation effort. For these reasons, fine-grained classification remains a formidable challenge for traditional deep neural networks (DNNs). This is primarily due to their limited capacity to discriminate between fine-grained features and the inherent difficulty in learning detailed patterns from limited training data.

This paper presents a **GNN P**ost-**H**oc (GPH) plugin that leverages the power of graph neural networks (GNNs) to enhance existing fine-grained image classification methods. We propose a design architecture that integrates GNN blocks into a conventional DNN architecture, allowing for the extraction of fine-grained features while maintaining the robustness and generalization capabilities of deep learning. Our approach aims to capture intricate inter-dependencies between feature vectors, effectively clustering them into meaningful groups that correspond to fine-grained categories. By doing so, we aim to improve the classification accuracy, particularly in scenarios where intra-class variations are significant.

In this work, we provide a comprehensive investigation into the effectiveness of our proposed model, benchmarking it against state-of-the-art methods on widely recognized fine-grained classification

datasets. We demonstrate that the incorporation of GNN blocks leads to substantial performance gains, showcasing the potential of this hybrid approach for fine-grained image classification tasks. Our contributions can be summarized as follows:

- We introduce a novel network architecture design in which GNN blocks are incorporated following the DNN encoder, improving the ability to cluster feature vectors and mitigating the ambiguity issue in fine-grained classification.

- The proposed design can be easily integrated into various fine-grained classifiers, enhancing performance, while the model's complexity and processing time remain manageable.

- Our extensive experiments on publicly available datasets demonstrate the model's capability to enhance feature clustering and accuracy, while also achieving state-of-the-art results on the Stanford Dogs dataset.

The remainder of this paper is organized as follows: Section 2 provides an overview of related work in the field of fine-grained classification and graph neural networks. In Section 3, we present our proposed model architecture in detail. Section 4 describes the experimental setup and presents empirical results. Finally, in Section 5, we discuss the implications of our findings and outline avenues for future research.

## 2 RELATED WORK

In this section, we present two research tracks related to our study, including fine-grained image classification and graph neural networks.

### 2.1 FINE-GRAINED IMAGE CLASSIFICATION

Recent deep learning research on fine-grained classification problems has primarily focused on two main directions, including convolutional neural networks (CNN)-based methods and visual attention-based methods.

**CNN-based Fine-Grained Image Classification** is commonly seen in general classification tasks and specifically in fine-grained classification problems. Common baseline CNN architectures such as MobileNet Howard et al. (2017), DenseNet Huang et al. (2017), ConvNeXT Liu et al. (2022), and others can also be applied to fine-grained classification tasks. Notably, in 2022, both task-specific models the PIM (Chou et al., 2022) and the $\mu$2Net+ (Gesmundo, 2022) achieved state-of-the-art performance on the NABirds and CUB-200-2011 datasets (Wah et al., 2011). Currently, the HERBS model (Chou et al., 2023) stands out as one of the top-performing models on these datasets. It employs two innovative approaches, namely high-temperature refinement and background suppression, to address key challenges in fine-grained classification.

**Visual attention-based approaches** aim to mimic human visual attention by selectively focusing on informative regions or features within an image. One of the pioneering models utilizing this mechanism, Xiao et al. (2014), uses two-level attention to concentrate on both overall image context and fine-grained details. More recently, a reinforcement learning-based fully convolutional attention localization network (Liu et al., 2017) adaptively selects multiple task-driven visual attention regions. This model is renowned for being significantly more computationally effective in both the training and testing phases. Furthermore, the ViT-NeT (Kim et al., 2022) model augments the explicability of Vision Transformers (Dosovitskiy et al., 2021) by integrating a neural tree decoder, enabling the generation of predictions with hierarchical structures that facilitate improved comprehension and examination of the model's decision-making process. In another context, MetaFormer Yu et al. (2021) employs convolutional layers to encode visual information and transformer layers to fuse vision and meta information. Currently, the ViT-NeT and MetaFormer models are achieving the highest accuracy levels on the Stanford Dogs dataset(Khosla et al., 2011), and the NABirds dataset Van Horn et al. (2015), respectively.

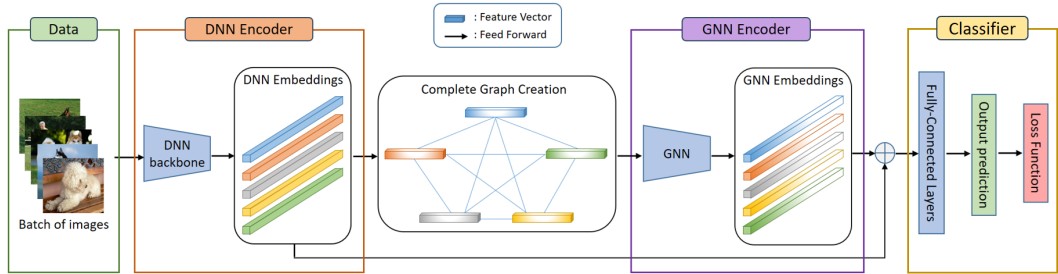

Figure 1: An overview of our GNN-based post-hoc architecture design

## 2.2 GRAPH NEURAL NETWORKS

Graph neural networks (GNNs) can be categorized into four types, which encompass: convolutional graph neural networks (ConvGNNs), recurrent graph neural networks (RecGNNs), graph autoencoders (GAEs), and spatial-temporal graph neural networks (STGNNs). Inspired by the success of CNNs in computer vision, numerous methods have emerged to redefine convolution for graph data. These methods, collectively known as Convolutional Graph Neural Networks (ConvGNNs), can be categorized into two main streams: spectral-based and spatial-based approaches. Since the pioneering work on spectral-based ConvGNNs was presented by Bruna et al. (2014); various advancements, extensions, and approximations have been made in spectral-based ConvGNNs including GCN Kipf & Welling (2017), AGCN Li et al. (2018), and DualGCN Zhuang & Ma (2018). On the other hand, Spatial-based ConvGNNs define graph convolutions based on a node's spatial relations (e.g., Veličković et al. (2019); Xu et al. (2019); Chiang et al. (2019)). From a different perspective, spatial-based ConvGNNs share a similar concept of information propagation and message passing with RecGNNs. Furthermore, alongside RecGNNs and ConvGNNs, several other GNN variants have been devised, including Graph Autoencoders (GAEs) Kipf & Welling (2016) and Spatial-Temporal Graph Neural Networks (STGNNs) Yu et al. (2018).

## 3 PROPOSED APPROACH

### 3.1 PROBLEM DEFINITION

For the problem of fine-grained image classification, similar to the general image recognition, we are given a training dataset $\mathcal{T} = \{(x_i, y_i)\}_{i=1}^N$ drawn from an unknown joint data distribution defined on $\mathcal{X} \times \mathcal{Y}$, with $\mathcal{X} \subset \mathbb{R}^{3 \times H \times W}$ and $\mathcal{Y} \subset \{0, 1\}^C$ denoting the input image space and the output label space ($H, W$ denoted as height and width of an image in $\mathcal{X}$). In particular, the label space $\mathcal{Y}$ - which contains one-hot classification vectors, is the union space of all the $C$ subspaces corresponding to the $C$ subordinate categories of the same meta-category, i.e., $\mathcal{Y} = \mathcal{Y}_1 \cup \mathcal{Y}_2 \cup \cdots \cup \mathcal{Y}_c \cup \cdots \cup \mathcal{Y}_C$. Our goal is to learn a mapping function $f : \mathcal{X} \rightarrow \mathcal{Y}$ that correctly classifies images into one of the $C$ categories.

### 3.2 GPH ARCHITECTURE DESIGN

In order to improve the model's understanding of complex image relationships and bolster its capability to distinguish subtle variations in fine-grained classification tasks, we propose a *simple yet effective* architecture design that utilizes a plug-in module based on GNNs. Figure 1 illustrates the workflow of our proposed design, in which the GNN encoder can be considered as a post-hoc plug-in. We first utilize a DNN-based encoder to generate feature vectors. These vectors are then constructed into a complete graph and input into a GNN model to obtain GNN embeddings, aiming to enhance the discriminative ability between feature clusters. The two features from the two encoders are then combined and fed into fully connected layers for classification. It is worth noting that the GNN plug-in can be integrated into any mainstream backbone network such as DenseNet, Swin Transformer, and ConvNeXT. In this section, we offer comprehensive insights into our GNN Post-Hoc structure, consisting of two primary components: the deep neural network encoder and the graph neural network encoder, along with an overview of the inference process.

In our network architecture, function $f$ consists of three components: (1) a deep neural network encoder $\Phi : \mathcal{X} \to \mathbb{R}^m$ that maps each input image $x_i$ to a $l_2$-normalized feature embedding $z_i$; (2) a graph neural network encoder that constructs a fully connected graph $\mathcal{G}$ from the obtained feature vectors within a batch $\mathbf{z} = \{z_i\}_{i=1}^b$ and then maps them to $l_2$-normalized feature embeddings $\mathbf{g} = \{g_i\}_{i=1}^b$ with $g_i \in \mathbb{R}^m$; (3) a classifier $\Psi : \mathbb{R}^m \to \mathbb{R}^C$ that maps each feature in the combined $m$-dimensional embeddings of $z$ and $g$ to a classification vector, where a cross-entropy loss can be applied after using a sigmoid function.

**DNN encoder.** This encoder can be a typical encoder in any DNN-based image classification methods. Given a training batch $\{x_i, y_i\}_{i=1}^b$ with batch size $b$, the images are fed into the feature extractor network, yielding $l_2$-normalized embeddings $\{z_i\}_{i=1}^b$: $z_i = \Phi(x_i)$

**GNN encoder.** We enhance the capability of conventional classification networks for fine-grained classification tasks by incorporating a graph neural network module after their feature extraction module. Figure 2 illustrates a toy example depicting the distribution of feature points corresponding to images in a two-dimensional space. In Figure 2 (a), the features extracted by conventional models exhibit good class separability, with features from the same class clustering closely together. However, there is a lack of clear differentiation between clusters of different classes, leading to potential misclassifications. On the other hand, our model also facilitates the grouping of elements of the same class while improving the separation between clusters of different classes, thereby enhancing the overall accuracy.

We denote a fully connected graph $\mathcal{G} = (\mathcal{V}, \mathcal{E}, \mathcal{F})$, where $\mathcal{V}$ represents the set of images in each batch, i.e., $|\mathcal{V}| = b$, $\mathcal{E} = \{e_{ij}\}_{i,j=\overline{1,b}}$ is the set of edges connecting images, and $\mathcal{F} = \{z_1, z_2, ..., z_b\}$ is the node features in the graph.

Our proposed GPH can employ various GNN architectures as the GNN encoder, such as GraphTransformer Yun et al. (2019) and Graph-SAGE Hamilton et al. (2017) to learn the node embeddings, which are described by the feature matrix in $Z \in \mathbb{R}^{b \times m}$. Specifically, the initial node representation, which is the set of DNN embeddings $\{z_i\}_{i=1}^b$, are passed through multiple layers, with each layer encompassing two critical functions: AGGREGATE, responsible for gathering information from the neighbors of each node, and COMBINE, tasked with updating the node representations by combining the aggregated information from neighbors with the current node representations.

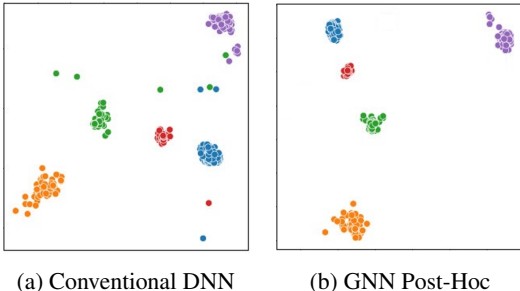

(a) Conventional DNN     (b) GNN Post-Hoc

Figure 2: Example of feature embeddings of GPH

Mathematically, the general framework of the GNN encoder can be expressed as follows:

- Initialization: $Z^{(0)} = \mathcal{F}$.
- For each layer $l$-th of the GNN encoder ($l = \overline{1, L}$ with $L$ is the number of layers), we update the embeddings of the graph to have $Z^{(l)} = \{z_i^{(l)}\}_{i=1}^b$, which is encoded through two general functions (here, $z_i^{(0)}$ refers to $z_i$):

$$a_i^{(l)} = \mathbf{AGGREGATE}^{(l)}\left\{z_j^{(l-1)} : j \in \mathcal{N}(i)\right\}, \quad z_i^{(l)} = \mathbf{COMBINE}^{(l)}\left\{z_i^{(l-1)}, a_i^{(l)}\right\} \tag{1}$$

where $\mathcal{N}(i)$ is the set of neighbors for the $i$-th node.

**Feature combination.** The node representations $Z^{(L)}$ obtained at the last layer of the GNN encoder can be treated as the final node representations, and these features are subsequently merged with features from the DNN encoder $\{z_i\}_{i=1}^b$ as follows:

$$c_i = \mathbf{COMBINE}\left\{z_i^{(L)}, z_i\right\}. \tag{2}$$

These final features $\{c_i\}_{i=1}^b$ are then passed through the classifier $\Psi$ for classification.

Table 1: Dataset statistics. Imbalance is defined as the ratio of the number of images in the largest class to the number of images in the smallest class.

| Dataset | # Train | # Test | Imbalance |
|---|---|---|---|
| CUB-200-2011 | 5,994 | 5,794 | 1.03 |
| Stanford Dogs | 12,000 | 8,580 | 1.00 |
| NABirds | 23,929 | 24,633 | 15.00 |

## 4 EXPERIMENTS

### 4.1 DATASETS AND EXPERIMENTAL SETTINGS

**Datasets.** We perform experiments on three well-known fine-grained datasets: CUB-200-201 Wah et al. (2011), Stanford Dogs Khosla et al. (2011), and NABirds Van Horn et al. (2015). First, the CUB-200-201 dataset, i.e., Caltech-UCSD Birds-200-201, comprises 11,788 labeled images of bird species. Originally, the dataset included 200 bird species, but the extended version incorporates extra images for each category, resulting in a total of 201 classes. This dataset also provides attribute labels and landmark annotations, which offer supplementary information for detailed analysis. Second, the Stanford Dogs dataset consists of 20,580 images featuring 120 distinct dog breeds, and it does not include meta-information similar to CUB-200-201. And finally, the NABirds dataset, short for "North American Birds Dataset," contains over 48,000 annotated images of 555 bird species found in North America. The division of training and testing data follows the predefined configurations in each dataset, with detailed quantities provided in Table 1.

**Implementation details.** All experiments are conducted on an NVIDIA Tesla T4 GPU with 15GB of RAM. Initially, all input images are resized to 224x224 pixels. We employ simple data augmentation techniques such as RandomHorizontalFlip and RandomRotation during training. The DNN encoder is trained using pre-trained weights from the ImageNet1K dataset. For the GNN encoder, we integrate four blocks in total. The first block transforms the output features of the base encoder into embeddings with a size of 1024. The remaining three blocks further transform the features to ensure that the output features have a consistent dimension of 1024. The model is fine-tuned for 50 epochs using a batch size of 32 for all models. As the proposed GPH can be influenced by the batch size, we provide detailed experiments to evaluate the results corresponding to different batch size configurations in section 4.2.3. We train the network using the Rectified Adam optimizer with a default epsilon value of $1e^{-8}$. The dimension of the embedding of the encoder network is set to 1024. We evaluate the top-1 classification error on the shuffled validation set. Additionally, the initial learning rate is set to $1e^{-5}$.[1]

### 4.2 EXPERIMENTAL RESULTS

Our empirical studies in this subsection are designed to answer the following key research questions.

- **Q1.** How is the effectiveness of the proposed design when applying various types of GNN encoders to the GPH architecture?
- **Q2.** To what extent does the GNN Post-Hoc model improve performance compared to regular classification networks and state-of-the-art fine-grained classification approaches?
- **Q3.** How do batch configurations affect the performance of the proposed model?
- **Q4.** How does integrating an additional GNN encoder with the DNN encoder impact the representation of feature vectors compared to a conventional classification model?
- **Q5.** How does the GNN aggregation functions affect the accuracy of the proposed model?

### 4.2.1 DIFFERENT GNN ENCODERS (Q1)

To investigate the effect of employing different GNN models as the GNN encoder, we perform an experiment using four popular GNN methods, including: GCN Kipf & Welling (2017), GAT

---

[1]The source code of the implementation is available online (currently omitted due to blind review).

Veličković et al. (2017), GraphSAGE Hamilton et al. (2017), and GraphTransformer Yun et al. (2019), by assessing their performance on the three benchmark datasets while using Densenet201 as the underlying DNN backbone. Furthermore, we introduce another baseline plug-in adopting an Attention layer instead of the GNN encoder for comparison against the GPH architecture. Table 2 reveals that models equipped with these additional modules consistently enhance accuracy in contrast to the standard Densenet201. Remarkably, our four GPH models exhibit even more substantial improvements, particularly in the context of fine-grained classification across these three datasets.

Table 2: Model accuracy according to different GNN encoders.

| Model | Acc (%) | | |
|---|---|---|---|
| | Stanford Dogs | CUB-200-2011 | NAbirds |
| Densenet201 | 83.95 | 79.13 | 77.55 |
| Densenet201-Attention | 85.28 | 79.45 | 78.59 |
| Densenet201-GCN | 87.6 | 84.40 | 84.14 |
| Densenet201-GAT | 87.82 | 84.61 | 83.94 |
| Densenet201-SAGE | 87.39 | 84.43 | 83.54 |
| Densenet201-GraphTransformer | 88.09 | 84.48 | 83.62 |

### 4.2.2 COMPARISON WITH EXISTING METHODS (Q2)

**Baselines**. To validate the effectiveness and generalization of our method, we investigate the performance of incorporating GPH on four different well-known DNNs and their variants, including DenseNet Huang et al. (2017), MobileNet Howard et al. (2017), ConvNext Liu et al. (2022), and SwinTransformer Liu et al. (2021), HERB Chou et al. (2023). It is important to highlight that our GPH is the only modification, while all other training configurations and hyperparameters remain unaltered from the original implementations. For consistency, we employ GraphTransformer as the GNN encoder for all experiments in this section. Even though we incorporate our proposed method across various techniques and assess it on diverse datasets, we maintain the consistent parameter configuration detailed earlier throughout all experiments.

**Comparison results.** Table 3 shows the impact of our GPH on fine-grained classification performance across different methods and datasets. Our interesting findings are summarized as follows:

- The table clearly illustrates that the incorporation of GPH consistently improves fine-grained classification results. Notably, we observe an average increase of $+2.78\%$, $+3.83\%$, and $+3.29\%$ on the Stanford Dogs, CUB-200-2011 datasets and NABirds, respectively.

- While GPH significantly enhances the performance of CNN-based models on both datasets, the improvement is more moderate for transformer-based models. We hypothesize that because of the inherent similarity between the attention mechanism of transformers and the nature of GNN, the accuracy improvement is not as substantial as with CNN-based models. For example, with models like DenseNet and MobileNet, accuracy increases by $3 - 6\%$ on both datasets, while with Swin Transformer, it ranges from $1 - 2\%$. Notably, ConvNext shows a slight performance boost on the Stanford Dogs dataset but a significant improvement of $5 - 6\%$ on CUB-200-2011.

- Improving existing fine-grained classification methods is a challenging endeavor. However, as shown in Table 3, our proposed approach achieves new state-of-the-art results on the Stanford Dogs dataset[2]. It is worth noting that for the other two datasets, including CUB-200-2011 and NAbirds, we fail to reproduce the performance of state-of-the-art baselines, i.e., HERB[3] and MetaFormer[4] even when referring to their GitHub pages.

- Additionally, we observe that for some models, when we add the GPH module to smaller variants, they achieve better accuracy than the larger variants without the module, while also being less time-consuming and complex. For instance, SwinT-Small-GPH (61.7M

---

[2]According to the comparison table in `https://paperswithcode.com/sota/fine-grained-image-classification-on-stanford-1` on 26/09/2023.

[3]`https://github.com/chou141253/FGVC-HERBS.git`

[4]`https://github.com/dqshuai/MetaFormer.git`

Table 3: The impact of GPH on fine-grained classification outcomes when incorporated into various DNN techniques. The accuracy gain when applying GPH is provided in the brackets.

| Method | Inference time | # params | Acc (%) | | |
|---|---|---|---|---|---|
| | | | Stanford Dogs | CUB-200-2011 | NABirds |
| MobilenetV3-S | 0.013 | 1.6M | 73.12 | 67.5 | 66.46 |
| MobilenetV3-S-GPH | 0.016 | 17.4M | 77.01(+3.89) | 69.86(+2.36) | 69.1(+2.64) |
| MobilenetV3-L | 0.035 | 4.4M | 78.31 | 77.65 | 75.86 |
| MobilenetV3-L-GPH | 0.039 | 23.2M | 82.72(+4.41) | 80.77(+3.12) | 79.82(+3.96) |
| Densenet201 | 0.28 | 18.3M | 83.95 | 79.13 | 77.55 |
| Densenet201-GPH | 0.29 | 73.7M | 87.72(+3.77) | 84.48(+5.35) | 83.81(+6.26) |
| Densenet161 | 0.42 | 26.7M | 84.46 | 79.68 | 78.97 |
| Densenet161-GPH | 0.45 | 88.7M | 88.47(+4.01) | 84.79(+5.11) | 84.75(+5.78) |
| SwinT-Small | 0.51 | 49.1M | 91.39 | 86.27 | 86.74 |
| SwinT-Small-GPH | 0.52 | 61.7M | 92.79(+1.40) | 87.35(+1.08) | 87.97(+1.23) |
| SwinT-Big | 0.82 | 87.0M | 92.11 | 85.86 | 86.32 |
| SwinT-Big-GPH | 0.84 | 102.8M | 93.06(+0.95) | 87.9(+2.04) | 88.03(+1.71) |
| ConvNextBase | 0.59 | 88.7M | 92.77 | 81.93 | 85.31 |
| ConvNextBase-GPH | 0.61 | 103.4M | 94.56(+1.79) | 87.52(+5.59) | 87.86(+2.55) |
| ConvNextLarge | 1.22 | 197.9M | 93.71 | 81.74 | 85.53 |
| ConvNextLarge-GPH | 1.23 | 231.8M | 95.79(+2.08) | 87.8(+6.06) | 88.11(+2.58) |
| HERB-SwinT | 1.74 | 286.6M | 88.62 | 89.9 | 90 |
| HERB-SwinT-GPH | 1.88 | 318.2M | 88.9(+0.28) | 90.37(+0.47) | 90.61(+0.61) |
| **Avg. Improvement** | | | +2.51 | +3.46 | +3.04 |

parameters) outperforms SwinT-Big (87M parameters), and ConvNextBase-GPH (103.4M parameters) surpasses ConvNextLarge (197.9M). This partly demonstrates the effectiveness of the proposed module when integrated into different backbones. Regarding neural network complexity, despite a significant increase in the number of parameters in the proposed models compared to the base ones, the inference time varies only slightly between them.

In summary, our proposed approach consistently demonstrates enhanced performance across various classifiers and fine-grained datasets. Moreover, our method can easily integrate with cutting-edge classifiers to yield further enhancements. Notably, the parameter configuration for our approach remains uncomplicated, delivering favorable outcomes with a single setup across diverse classifiers and datasets.

### 4.2.3 THE IMPACT OF BATCH CONFIGURATIONS (Q3)

In both the training and inference phases of the proposed module, the feature learning process of the GNN encoder begins by constructing a complete graph based on the features of the DNN encoder within a batch. Therefore, batch configurations, including batch size and how images are selected, influence the model's performance to some extent. In this part, we will examine the stability of GPH under different batch configurations.

**Batch size**. Figure 3 reveals that altering the batch size of the training and testing process has minimal impact on the accuracy of the baseline DNN models. Therefore, in this experiment, we only compare the results of 4 out of the 9 GPH variants for ease of illustration. The results plotted on both datasets demonstrate that larger models tend to exhibit higher stability, i.e., changes in batch size do not significantly affect performance. Among the models, Densenet201-Attention and MobileNet exhibit the biggest variability. In contrast, the other 3 models show differences of less than 1%.

**Shuffling the validation dataset during evaluation.** Since GPH refines image latent embeddings using a fully connected graph of all embeddings within a batch, its performance may depend on the variation of the samples in the batch. In this section, we examine the stability of GPH under different batch configs of the evaluation datasets. Table 4 displays the comparison results of Densenet161-GPH and SwinT-Big-GPH models on the validation dataset with two different orders: sequential and shuffled-data sampling. In the sequential data sampling scenario, data is drawn from one class

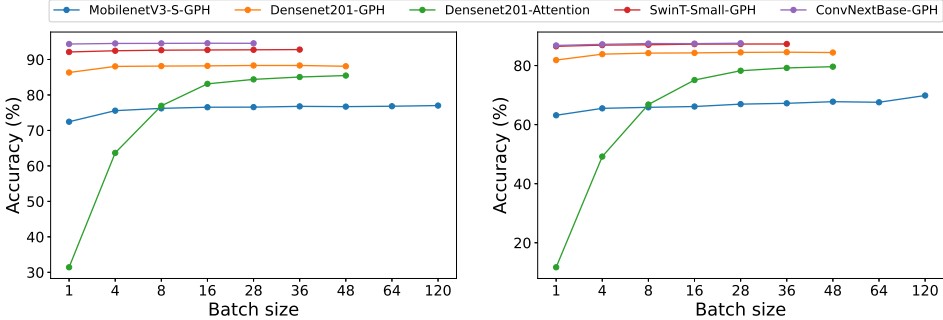

Figure 3: Performance comparison for GPHs using various batch sizes on both the Stanford Dogs dataset (on the left) and the CUB-200-2011 dataset (on the right). Note that experiments with large batch sizes on Densenet201-GPH, SwinT-Small-GPH, and ConvNextBase-GPH are omitted due to the GPU's memory constraints.

Table 4: Evaluation results on the three datasets employing two distinct data sampling techniques during validation, namely Sequential and Shuffle.

| Method | Stanford Dogs | | CUB-200-2011 | | NABirds | |
|---|---|---|---|---|---|---|
| | Sequential | Shuffle | Sequential | Shuffle | Sequential | Shuffle |
| Densenet161-GPH | 88.47 | 88.17 | 84.79 | 84.53 | 84.75 | 84.62 |
| SwinT-Big-GPH | 93.06 | 92.82 | 87.90 | 87.66 | 87.38 | 87.21 |

before moving on to the next class when filling the batches, making the variation of samples within each batch low. In contrast, in the common shuffled-data sampling, the variation within each batch is high since each sample is randomly picked from any class. As reported in Table 4, sequential sampling provides slightly better accuracy, but the gap is small (maximum 0.3%). Therefore, we can confirm that GPH provides a pretty stable result, and the diversity of classes within the same batch has a minor impact on the model's classification performance.

**Feature selection within a batch during evaluation and prediction.** The question at hand is whether, with pre-trained weights obtained during the training of the GPH model and a batch size of $b$, the model's input during testing or inference must necessarily be fixed with $b$ images for the GNN encoder to process. To address this question, we employ a method of filling the batch embedding with vectors of all ones. Specifically, assuming we have $b_t < b$ images for testing, $b_t$ images first pass through the DNN encoder to extract features $\{z_i\}_{i=1}^{b_1}$. Then, the $\{z_j\}_{j=b_1}^{b}$ are initialized as vectors of ones, and the entire set of $b$ features is subsequently input into the GNN encoder for processing, as described in Section 3.2. Table 5 presents the evaluation results on the validation set using this method with $b_1 = 1$, corresponding to different values of $b$. The results demonstrate the stability of the batch size-specific filling method, even with MobileNetV3-S-GPH, where this method achieves better accuracy than the conventional approach of taking the entire batch of images. Notably, the results for Densenet201-Attention-filled are favorable, while Densenet201-Attention performs poorly with a small batch size. From these results, it is evident that the filling method effectively addresses the posed question.

Table 5: The performance of models using various batch sizes after filling batch feature embeddings with ones tensors on the Stanford Dogs dataset.

| | 1 | 4 | 8 | 16 | 28 | 36 | 48 | 64 | 120 |
|---|---|---|---|---|---|---|---|---|---|
| MobilenetV3-S-GPH | 72.54 | 75.57 | 76.22 | 76.54 | 76.57 | 76.78 | 76.71 | 76.82 | 77.01 |
| MobilenetV3-S-GPH-filled | 76.2 | 76.33 | 76.52 | 76.64 | 76.93 | 76.98 | 76.94 | 76.92 | 77.01 |
| Densenet201-Attention | 31.42 | 63.67 | 76.94 | 83.13 | 84.39 | 85.06 | 85.47 | _ | _ |
| Densenet201-Attention-filled | 85.32 | 85.61 | 85.63 | 85.56 | 85.6 | 85.64 | 85.47 | _ | _ |
| Densenet201-GPH | 86.33 | 88.05 | 88.14 | 88.2 | 88.31 | 88.31 | 88.09 | _ | _ |
| Densenet201-GPH-filled | 87.25 | 87.47 | 87.68 | 88 | 88.27 | 88.27 | 88.09 | _ | _ |

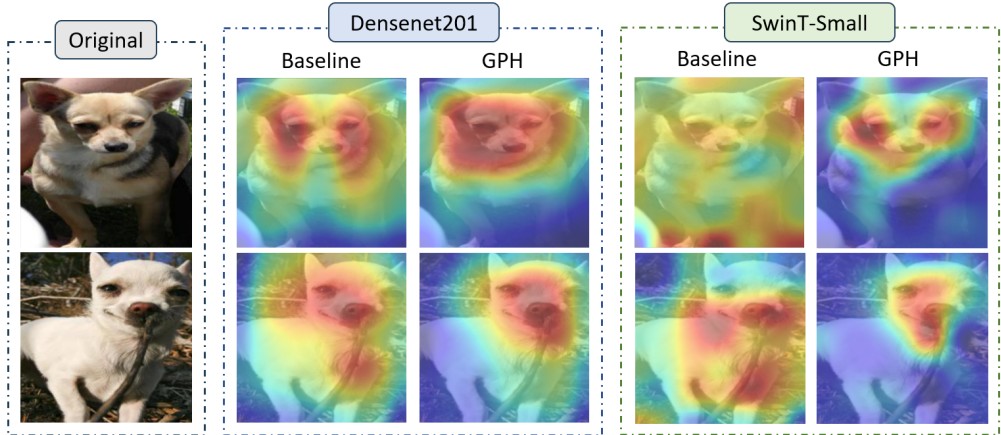

Figure 4: The visualization of the attention map between image feature maps and the four models

### 4.2.4 VISUAL ANALYSIS (Q4)

To identify the areas of primary interest in the images according to the models' analysis, we utilize Grad-Cam Selvaraju et al. (2019) to display their activation maps on the original images, as depicted in Figure 4, where the color spectrum from blue to red represents values from low to high, with higher values indicate stronger focus of the model on that area. We can discern that all four models primarily concentrate on the object in the image, which is the dog. Nevertheless, in the case of the Densenet201-GPH and SwinT-Small-GPH models, our model gives more attention to the dog's facial regions, seeking cues for assessment, whereas the baseline's heatmap weights are spread across the entire dog.

### 4.2.5 GNN AGGREGATION FUNCTIONS (Q5)

Table 6: The impact of GNN aggregation functions on fine-grained classification

| Model | Sum | Mean |
|---|---|---|
| Densenet201-Attention | 75.15 | 85.85 |
| Densenet201-SAGE | 66.90 | 87.39 |

The results presented in Table 6 illustrate a comparison between various GNN aggregation functions, specifically SUM and MEAN. These two functions yield divergent impacts on accuracy. The MEAN function leads to a notable improvement in model accuracy compared to DenseNet201, whereas the SUM operation has a detrimental effect. Additionally, we observe contrast in performance: when using the SUM function, SAGE achieves a lower accuracy than Attention, while the opposite holds true for the MEAN function.

## 5 CONCLUSION AND DISCUSSIONS

In our investigation, we identified a novel architectural design that appears deceptively straightforward yet has remained unexplored in prior studies. Rigorous experimentation conducted on benchmark datasets underscores the efficacy of our proposed approach, showcasing its seamless integration with a variety of fine-grained classifiers. These synergistic interactions yielded appreciable improvements in accuracy, establishing a new benchmark for performance in the field. Additionally, our architectural innovation fostered a reduction in both model parameters and inference latency when compared to conventional DNN methodologies.

Our research opens up several promising avenues for future exploration. First, further investigation can delve into optimizing the architecture and hyperparameters of the integrated GNN-DNN model for different fine-grained classification tasks. Additionally, exploring different graph construction strategies and graph neural network architectures may yield insights into improving model performance. Moreover, the application of this integrated approach to other computer vision tasks and datasets warrants exploration, as it has the potential to enhance various aspects of visual recognition.

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

## APPENDIX

## A ALGORITHM FOR GNN POST-HOC

In this section, we present the previously omitted pseudo-code for the GPH model. Algorithm 1 outlines the common steps for constructing the GNN encoder architecture, which includes the GN-NModule followed by a Linear layer, ReLU, and Batch Normalization, acting as a block. The first block transforms the DNN features into new features with dimensions defined in the initialization of GNNEncoder ($embd\_size$). Subsequent blocks transform the features while maintaining the same dimensions. Algorithm 2 describes the process of adding the GNN encoder as an add-on to the DNN encoder.

**Algorithm 1** A GNN Encoder Network Pseudo-code, PyTorch-like

```python
# feat_size:  input feature size for GNN Encoder
# embd_size:  output embedding size of GNN Encoder
# n_layers:  number of GNN Encoder blocks
# n_cls:  number of classes
# edge_index:
class GNNEncoder(torch.nn.Module):
  def __init__(self, feat_size, embd_size, n_layers, n_cls):
    self.n_layers = n_layers
    self.gnn1 = GNNModule(feat_size, embd_size)
    self.gnns = ModuleList(GNNModule(embd_size, embd_size), n_layers)
    self.linear1 = Linear(embd_size, embd_size)
    self.linears = ModuleList(Linear(embd_size, embd_size), n_layers)
    self.lin_skip1 = Linear(feat_size, embd_size)
    self.lin_skips = ModuleList(Linear(feat_size, embd_size), n_layers)
    self.bn1 = BatchNorm1d(embd_size)
    self.bns = ModuleList(BatchNorm1d(embd_size), n_layers)
    self.out = Linear(embd_size, n_cls)
  def forward(self, x, edge_index):
    x_skip = self.skip1(x)
    x = self.gnn1(x, edge_index)
    x += x_skip
    x = relu(self.linear1(x))
    x = self.bn1(x)
    for i in range(self.n_layers):
      x_skip = self.lin_skips[i](x)
      x = self.gnns[i](x, edge_index)
      x += x_skip
      x = relu(self.linears[i](x))
      x = self.bns[i](x)
    x = self.out(x)
    return x
```

**Algorithm 2** A GNN Post-Hoc Network Pseudo-code, PyTorch-like

```python
class GNNPostHoc(torch.nn.Module):
  def __init__(self, embd_size, n_layers, n_cls):
    self.base_enc = DNNEncoder()
    feat_size = self.base_enc.classifier.in_features
    self.gnn_enc = GNNEncoder(feat_size, embd_size, n_layers, n_cls)
  def get_edge_index(batch_size):
    #
    return edge_index
  def forward(self, images):
    batch_size = images.shape[0]
    x = self.base_enc(images)
    edge_index = self.get_edge_index(batch_size)
    return self.gnn_enc(x, edge_index)
```

# B ADDITIONAL RESULTS

In Tables 7, 8, and 9, we provide a detailed results table for the GPH models on three datasets, including Stanford Dogs, CUB-200-2011, and NABirds. The experiments are conducted with batch size values in $\{1, 4, 8, 16, 28, 36, 48, 64, 120\}$. It is worth noting that the maximum batch size for each model is described specifically in Table 10 under the configuration of an NVIDIA Tesla T4 GPU with 15GB of RAM. These results further substantiate our observations in Research Question 3 regarding batch configurations.

Table 7: Standford Dogs.

|                    | 1     | 4     | 8     | 16    | 28    | 36    | 48    | 64    | 120   |
|--------------------|-------|-------|-------|-------|-------|-------|-------|-------|-------|
| MobilenetV3-S-GPH  | 72.54 | 75.57 | 76.22 | 76.54 | 76.57 | 76.78 | 76.71 | 76.82 | 77.01 |
| Densenet201-GPH    | 86.33 | 88.05 | 88.14 | 88.2  | 88.31 | 88.31 | 88.09 | –     | –     |
| SwinT-Small-GPH    | 92.12 | 92.46 | 92.62 | 92.68 | 92.73 | 92.79 | –     | –     | –     |
| ConvNextBase-GPH   | 94.36 | 94.52 | 94.54 | 94.58 | 94.56 | –     | –     | –     | –     |

Table 8: CUB-200-2011.

|                    | 1     | 4     | 8     | 16    | 28    | 36    | 48    | 64    | 120   |
|--------------------|-------|-------|-------|-------|-------|-------|-------|-------|-------|
| MobilenetV3-S-GPH  | 63.17 | 65.51 | 65.86 | 66.13 | 66.94 | 67.22 | 67.77 | 67.57 | 69.86 |
| Densenet201-GPH    | 81.86 | 83.86 | 84.22 | 84.29 | 84.46 | 84.53 | 84.41 | –     | –     |
| SwinT-Small-GPH    | 86.52 | 86.92 | 87.04 | 87.26 | 87.29 | 87.31 | –     | –     | –     |
| ConvNextBase-GPH   | 86.81 | 87.19 | 87.45 | 87.43 | 87.56 | –     | –     | –     | –     |

Table 9: NABirds.

|                    | 1     | 4     | 8     | 16    | 28    | 36    | 48    | 64    | 120   |
|--------------------|-------|-------|-------|-------|-------|-------|-------|-------|-------|
| MobilenetV3-S-GPH  | 59.72 | 66.05 | 67.27 | 68.05 | 68.4  | 68.64 | 68.77 | 68.83 | 69.1  |
| Densenet201-GPH    | 73.96 | 80.23 | 82.14 | 83.09 | 83.47 | 83.56 | 83.62 | –     | –     |
| SwinT-Small-GPH    | 85.13 | 86.43 | 87.07 | 87.64 | 87.88 | 87.97 | –     | –     | –     |
| ConvNextBase-GPH   | 85.72 | 87.33 | 87.57 | 87.78 | 87.86 | –     | –     | –     | –     |

Table 10: The maximum batch size for each model.

| Full RAM | |
|----------|------------|
| MODEL    | BATCH SIZE |
| MobilenetV3 | 120 |
| Densenet | 48 |
| SwinTransformer | 36 |
| ConvNext | 28 |
| HERB-SwinT | 5 |

