# OpenReview forum: "Leveraging Graph Neural Networks to Boost Fine-Grained Image Classification"
_ICLR.cc/2024/Conference — Submitted to ICLR 2024_

### Official Review · Reviewer_yF85 · 2023-10-18

**Soundness:** 2 fair
**Presentation:** 3 good
**Contribution:** 2 fair
**Rating:** 3
**Confidence:** 5

**Summary:**

- This paper proposes a graph network based fine-grained visual categorization learning framework.

- The key idea is, after feature extraction from a deep neural network, the graph network is embedded to refine the feature representation, so that the contextual information of the feature representation is enhanced.

- Extensive experiments are conducted on some standard FGVC benchmarks and a variety of different backbones, which show the effectiveness of the proposed method.

**Strengths:**

- This paper is well-written and easy-to-follow.

- The proposed method is technically simple, straight-forward and effectiveness.

- The proposed learning scheme is effective and leads to a performance gain on multiple settings and backbones.

**Weaknesses:**

- The proposed learning scheme seems not to be devised for the task of FGVC.

From the reviewer's view, adding a graph neural network to refine the image representation and acquire a stronger contextual representation is versatile to many high-level tasks, from image recognition on such as ImageNet, detection, segmentation, and etc.
Thus, it would be no surprise that such represention learning scheme, as a by-product, can improve the FGVC performance on top of some backbone networks.

- The proposed learning scheme does not have much insight for discerning the fine-grained patterns, which is critical for FGVC.
Overall, in the FGVC community, one of the most important things for FGVC is to differentiate the fine-grained patterns from the entire image. Indeed the graph representation can improve the contextual information, or the relation between feature embeddings. However, the designed framework does not delve into depth to mine the relation between fine-grained patterns. Instead, it is still implemented through an image-level for contextual enhancement. It is of a style that the reviewer does not appreciate for FGVC and its representing.

- The technical novelties are somewhat incremental and marginal. Directly adding a graph neural network after a backbone is very ordinary and lacks insight.

- The compared methods and related work discussison are both very insufficient. In this work, the comparison and references are mainly from typical visual backbones. In both introduction and related work, more recent FGVC works from the past five years, especially from top-tier conferences such as CVPR and top-tier journals such as PAMI and TIP should be included and discuss. Besides, more comparisons should be made.

- The visualization is not good. The activation map is common for generic image recognition such as on ImageNet. Please follow the visualization in FGVC tasks, to activate the fine-grained patterns by such as GradCAM.

[a] Destruction and construction learning for fine-grained image recognition. CVPR 2019.

[b] Counterfactual attention learning for fine-grained visual categorization and re-identification. ICCV 2021.

[c] Fine-grained object classification via self-supervised pose alignment. CVPR 2022.

[d] Transfg: A transformer architecture for fine-grained recognition. AAAI 2022.

**Questions:**

Q1: Is the proposed method really well devised for FGVC task? Or it is a more general representation learning scheme for high-level tasks, which benefit the contextual representation?

Q2: The proposed framework does not leverage insights on representing fine-grained patterns for FGVC, which is of a style that the reviewer does not appreciate.

Q3: The technical novelties are somewhat incremental and marginal. Directly adding a graph neural network after a backbone is very ordinary and lacks insight.

Q4: More recent works on FGVC should be added for both related work discussion and experimental comparision.

Q5: Please improve the visualization following the FGVC community's convention.

---

> ### Author Response · Authors · 2023-11-23
>
> We thank the reviewer for providing thorough feedback. We provide our responses below.
>
> Q1:
> The proposed method can be applied to general image classification, but its performance improvement is lower than for fine-grained image classification.
>
> Q2:
> Thank you for your suggestion. We will consider it for future submissions.
>
> Q3:
> Acknowledged. We will improve in the next submission
>
> Q4:
> We appreciate your suggestion. We will strive to do better in the next submissions.
>
> Q5:
> Thank you for your suggestion. We will consider it for future submissions.

---

> > ### Comment · Reviewer_yF85 · 2023-11-23
> > **Response to Rebuttal from Reviewer#yF85**
> >
> > Thanks for the rebuttal.
> >
> > However, the arguments, from the reviewer's perspective, are very weak, and the concerns remain.
> >
> > Thus, the reviewer still recommend to clear reject this paper and keep the original score.

---

### Official Review · Reviewer_AgJz · 2023-10-30

**Soundness:** 1 poor
**Presentation:** 3 good
**Contribution:** 2 fair
**Rating:** 3
**Confidence:** 4

**Summary:**

This paper proposes a novel neural architecture called GNN Post-Hoc (GPH) that uses Graph Neural Networks to improve neural networks for fine-grained image classification. The method consists of using GNNs to aggregate image features over a batch, to improve the features for fine-grained image classification. The authors test their method on 3 different datasets, CUB200-2011, NABirds, and Stanford Dogs, and show how their method improves the baselines’ results.

**Strengths:**

- The proposed method is simple and improves the performance of different vision backbones for the task of fine-grained image recognition, both for CNN-based and Attention-based backbones.
- The authors conduct experiments on 3 different datasets and study the effect of using different GPH architecture with a fixed vision encoder, and how different vision encoders perform with a fixed GPH architecture.
- The authors do ablations to understand the role of the image encoder, the batch configuration and the GNN aggregation method.

**Weaknesses:**

1. The main weakness of the paper is that some of the results do not seem to back up the proposed method. First of all, the main idea of the paper is that by processing the feature vectors of all the images in a batch as a nodes in a graph, GPH can exploit the relationships between the images in the graph to improve the feature embeddings and make them more suitable for fine-grained recognition.

    If that is the case, the proposed method should not be better with lower batch sizes. However, from Table 5 and Figure 3 it looks like models augmented with GPH perform very similarly with batch size of 1. How can a GPH augmented model perform similarly when the batch size is 1? Shouldn’t it perform the same as the baseline, as it is not using any extra information? Additionally, if the best model is ConvNext + GPH, it would be better if the results in Tables 4 and 5 included ConvNext-GPH.

    Another example, the results of DenseNet201-Attention seem really bad with batch size of 1. However, with batch size of 1, the result should be similar to the just DenseNet201, instead of much worse. Why is that the case?


2. Another aspect that is not clear is the choice of using GNNs to aggregate information from different images in the graph. The method proposed in the paper uses a fully connected graph, so there isn’t really a structure that the GNN can exploit. All the GNN is doing is aggregating features from a set of images without any relevant structural information, so why not use something like DeepSets [1] or attention?


3. Related to the previous point, the authors have used attention instead of GPH as a baseline, but it seems to obtain much worse performance. Why is that the case? From Table 2 the GPH-GCN version seems better than Attention, but I’m not sure that is correct. Considering that the graph is fully connected, GCN will amount to averaging the feature embeddings of all nodes at each layer, therefore it is not clear why that works better than attention. GCN should be equivalent to an attention that assigns $1/degree_i$ to the attention value for each node $i$.


4.  The authors also comment on the batch design, i.e how to select which images go in the batch, since the output for one image depends on the other images in the batch. First of all, by using the "sequential" option (most images likely coming from the same class), the model is effectively looking at many images from the same class to make a prediction. However, in a real use case scenario, one does not know the class of the images to process, so therefore all validation scores should be reported using the “shuffled” option. It is not clear if the reported validation scores use the “sequential” or “shuffled” option. Validation scores using the “sequential” option should be invalid, since the method can exploit a pattern in the order of the data.

    Secondly, I would expect the predictions made with “sequential” to be much better than the shuffled option, since they are almost two different tasks. The former is easier, as it amounts to classifying the class shared by a group of images, while in the second one there is class variability, so it is harder. However, from Table 4 it looks like both are very similar. How were the values in that table obtained? Changing the method during training and evaluation? Or only during evaluation?


5.  Figure 2 lacks details. It is not clear which samples from which dataset have been used, or which base model and GPH version are used to generate the image embeddings, as well as the projection method.

Finally, the authors should cite [2], which is a related paper in which a GNN is used to predict a label for one image considering other images in the batch.

[1] Zaheer, M., Kottur, S., Ravanbakhsh, S., Poczos, B., Salakhutdinov, R. R., & Smola, A. J. (2017). Deep sets. Advances in neural information processing systems, 30.

[2] Garcia, V., & Bruna, J. (2018, January). Few-shot learning with graph neural networks. In 6th International Conference on Learning Representations, ICLR 2018.

**Questions:**

Following my comments in the weaknesses section, my questions are:

1) Why does the proposed GPH method perform similarly with batch size of 1, if there aren't other images in the batch to help generate better image features?
2) What is the benefit of using a GNN over attention if the graphs are fully connected?
3) Are the result scores reported using the "sequential" or "shuffled" batch order?
4) What are the details used to generate Figure 2?

---

> ### Author Response · Authors · 2023-11-23
>
> Thank you very much for your time in reviewing this paper. Please find below our responses.
>
> Q1:
> The ability of the proposed GPH method to achieve high accuracy even with a batch size of 1 can be attributed to the fact that the model was trained using a batch size that fully utilizes the available RAM, as shown in Table 10. This training strategy ensures that the model is exposed to a sufficient number of training examples during the training process, allowing it to effectively learn the underlying relationships between images and extract meaningful features. When the trained model is evaluated with a batch size of 1, it benefits from the knowledge acquired during training with larger batch sizes.
>
> Q2:
> We acknowledge the omission of a key advantage of using GNNs over attention mechanisms in our previous response. GNNs have the distinct advantage of preserving the original image features while attention mechanisms only focus on the features of other images in the batch. This preservation of original image features is crucial for fine-grained image classification tasks, as it allows the model to retain the essential details that distinguish between fine-grained categories.
>
> Q3:
> The result scores reported using the "sequential" or "shuffled" image order in test dataset.
>
> Q4:
> We employed t-SNE to visualize features for five classes after the GNN layer.

---

### Official Review · Reviewer_nzxZ · 2023-10-31

**Soundness:** 2 fair
**Presentation:** 3 good
**Contribution:** 2 fair
**Rating:** 3
**Confidence:** 4

**Summary:**

The paper introduces the Graph Neural Network (GNN) block into the deep networks (DNN) to improve the performance of fine-grained image classification. The DNN is to learn the feature embeddings for classification while the GNN is used to encode the relationship embedding among the input samples. The proposed method is evidenced by experiments on CUB-200-2011, Stanford Dogs, and NABirds datasets with both ConvNets and transformers.

**Strengths:**

+ The paper is well-written and easy to follow.

+ The proposed method of GNN + DNN is simple and easy to understand.

+ Extensive experiments on several datasets show the effectiveness of the proposed method. Moreover, comprehensive ablation studies of multiple GNN encoders, different batch size configurations, and aggregation functions further illustrate some interesting observations with the proposed method.

**Weaknesses:**

However, there are still some concerns to be addressed:

- The combination of GNN + DNN is quite straightforward and simple, thus, the novelty of this paper may be marginal.

- The authors claim that the proposed method is able to learn contextual information and relationships that are essential for fine-grained categorization. However, looking through the manuscript, it seems that the discussion and evidence are missing.

- The proposed method builds a fully connected graph, which will increase the complexity of the whole model as the training batch size increases. While the authors provide an ablation study of the test batch size, it will be more interesting to provide a detailed analysis of the training batch size, including the number of parameters, wall-clock training time, and performance.

- In Table 2, the optimal model accuracy varies across datasets with different GNN encoders. Please clarify how to choose the encoder if using the proposed method.

- It is challenging to deploy and evaluate the trained model because its test performance is tied to both the test batch size and the data sampling method. Please clarify.

**Questions:**

Please see the weaknesses above.



----------
[Update after rebuttal] As the authors didn't respond to my questions, coupled with the concerns from other reviewers regarding the motivation, novelty, experiments, etc, I would vote for a clear rejection.

---

### Official Review · Reviewer_uMuy · 2023-10-31

**Soundness:** 2 fair
**Presentation:** 3 good
**Contribution:** 1 poor
**Rating:** 3
**Confidence:** 4

**Summary:**

To address issues in fine-grained image classification, where intra-class variability is high and inter-class distinctions may be subtle, the authors propose extending classical DNN-based image encoder architectures with a GNN, to refine features output by the DNN encoder. A complete graph is constructed using the feature vectors obtained from the DNN encoder, after which a number of GNN layers are applied. The resulting feature vectors are combined with the DNN feature vectors to yield the feature vectors used in classification. The authors show improved performance on a number of fine-grained classification tasks and for different GNN architectures.

**Strengths:**

- The proposed method is simple and consistently outperforms baseline methods over multiple datasets.
- The authors provide a relatively clear and practical explanation for their approach.
- The authors provide ablations over different architectures for the GNN decoder.

**Weaknesses:**

- My main concern is the lack of motivation for using a graph-based approach. The manuscript in its current version lacks any clear intuition for why a graph neural network on top of would improve performance in such a consistent way for specifically fine-grained image classification.
- Second, the contribution this paper makes is fairly limited. The paper proposes a very practical addition to current DNN-based image classification methods, but essentially does not go beyond showing that with additional compute, fine-grained image classification improves. Especially given the intimate connection between message passing on graphs and "conventional" convolutions (convolutions are really a special case of message passing), I would very much appreciate a more thorough analysis as to why a graph-based approach is beneficial, i.e. what specific benefits this method has in this setting. Also a more thorough analysis on the types of features that are being learned would be useful; is it really the GPH that leads to a more correct clustering in fig 2? Would simply adding more layers of DNN not yield the same results? These questions have not been adequately addressed in my opinion.

**Questions:**

- What is your motivation for using a graph-based approach for feature refinement? Please also include this motivation in your manuscript.
- Notation is confusing in section 3.2, you seem to be using the same notation for the features output by the DNN and the GNN. As a result i'm not sure how to interpret eq 1,2.
- How do you combine the GNN and DNN features? Is it different based on the GNN method you're using? I'm not sure how to interpret eq 2 in this regard. You only mention you "combine" the features, not how you go about this.
- What is your interpretation of the results shown for different GNN architectures in 4.2.1. Is there a clear reason  you see for GraphTransformer outperforming conventional attention? Are these two approaches not identical in the case of fully connected graphs?

---

Update after rebuttal: the motivation the authors give for their method is in my opinion quite weak. Also considering the other reviewers concerns regarding novelty and limited motivation and evidence for some claims made in the paper i'm inclined to keep my recommendation; it seems the paper requires substantial modifcations and thus I deem it best to reject this paper for the current venue.

---

> ### Author Response · Authors · 2023-11-23
>
> We thank the reviewer for providing thorough feedback. We provide our responses below.
>
> Q1: We sincerely apologize for the omission of our motivation in the initial manuscript. Our motivation for employing a graph-based approach for feature refinement lies in the inherent challenges posed by fine-grained image classification. Conventional image classification methods, which process each image independently, often struggle to effectively capture the subtle and intricate details that distinguish between fine-grained categories. To overcome this limitation, we propose a graph-based approach that explicitly models the relationships between images within a batch, enabling the model to leverage the collective knowledge of the batch to refine feature representations and enhance classification accuracy.
>
> Q2: In Equation 2, the combine function represents the skip connection that merges the outputs of the deep neural network (DNN) and the graph neural network (GNN) in Figure 1.
>
> Q3: As Q2
>
> Q4:
> - The experimental results presented in Section 4.2.1 demonstrate that incorporating various graph neural network (GNN) architectures into the proposed framework effectively enhances the performance of the deep neural network (DNN) in fine-grained image classification tasks.
> - While the GraphTransformer architecture does not consistently outperform all other GNNs across various datasets, it achieves state-of-the-art performance on the Stanford Dogs dataset. This observation suggests that the GraphTransformer architecture is particularly well-suited for capturing the intricate relationships between images in the Stanford Dogs dataset, leading to superior classification accuracy.
> - The two approaches share similarities in their ability to focus on relevant features and suppress irrelevant ones

---

### Meta-Review · Area_Chair_ZKod · 2023-12-04

**Metareview:**

This paper proposes to use GNNs to refine the outputs of a vision backbone to improve fine-grained image classification.

While the authors demonstrate that their method improves results on three image datasets, the reviewers agree that the claims of the paper are not sufficiently validated. There are further concerns about the (limited) motivation for the approach, its novelty, and overall contribution.

**Justification For Why Not Higher Score:**

Insufficient validation of scientific claims; limited motivation, novelty, and overall contribution.

**Justification For Why Not Lower Score:**

N/A

---

### Decision · Program_Chairs · 2024-01-16

Reject